# Compton Scattering Imaging of Liquid Water in Porous Carbon-Based Materials

**Naruki Tsuji [1], Yoichiro Tsuji [2], Yoshiharu Uchimoto [3], Hideto Imai [4] and Yoshiharu Sakurai [1,*]**

[1] Japan Synchrotron Radiation Research Institute (JASRI), Hyogo 679-5198, Japan; ntsuji@spring8.or.jp
[2] Fuel Cell Cutting-Edge Research Center Technology Research Association (FC-Cubic TRA), Tokyo 135-0064, Japan; y-tsuji@fc-cubic.or.jp
[3] Graduate School of Human and Environmental Studies, Kyoto University, Kyoto 606-8501, Japan; uchimoto.yoshiharu.2n@kyoto-u.ac.jp
[4] Nissan Analysis and Research Center (NISSAN ARC), Kanagawa 237-0061, Japan; imai@nissan-arc.co.jp
\* Correspondence: sakurai@spring8.or.jp; Tel.: +81-791-58-0833

**Abstract:** Synchrotron-based Compton scattering imaging with intense high-energy X-rays allows the visualization of light element substances in an electrochemical device under an operando condition. In this study, we apply this imaging technique to a water-contained, porous carbon-based composite, which is used as a material for the gas diffusion layer in polymer electrolyte fuel cells. Analyses of the two-dimensional intensity images of Compton scattered X-rays provide the cross-sectional distributions of liquid water, as well as the depth dependency of the water content. In addition, the analyses reveal a significant interaction between the carbon materials and water droplets.

**Keywords:** X-ray Compton scattering; porous carbon materials; fuel cells; cross-sectional imaging; high-energy X-rays

## 1. Introduction

PEFC (Polymer Electrolyte Fuel Cell) technologies are drawing much attention due to their clean emission, high power density and low temperature operation [1]. These attractive features have made PEFCs a promising candidate for the next-generation power source of transportation and portable applications. Although the need for developing high-performance PEFCs is rapidly increasing, many fundamentals remain to be clarified for further advancement. In PEFCs, the proton conductivity of the polymer electrolyte membranes depends on the water content, and excessive liquid water hinders electrochemical reactions at catalyst sites. Therefore, water management is crucially important for stable PEFC operation [2–4]. The liquid water generation and transport are generally inhomogeneous and random since the component materials are complex and porous. The liquid water content and its inhomogeneous distributions have been reported by neutron radiography [5–10] and X-ray computed tomography (CT) [11–15]. This experimental information is indispensable to simulation and modelling of PEFCs.

Although synchrotron-based X-ray CT has been used to probe the liquid water distribution, this technique requires the use of custom-made cells. This technique also requires sample rotation to obtain three-dimensional images. Neutron radiography has been used for non-destructive observation of water distributions in commercial cells. However, it is difficult to distinguish the local parts within the cell because of the limited spatial resolution of neutron beams. Compton scattering imaging (CSI) is unique among the synchrotron X-ray techniques, since it probes electron density in a local gauge volume. The CSI technique uses X-rays of 100 keV and higher, enabling non-destructive observation of light element substances inside electrochemical devices.

The synchrotron-based CSI technique has been developed to probe the inside of commercial lithium-ion batteries, and has successfully visualized the lithium-ion migration

in a commercial battery under discharge and the structural change due to the volume expansion [16]. In addition, a new analytical technique was developed using the line shapes of Compton scattered X-rays (S-parameter) [17]. The S-parameter analysis has successfully revealed the Li concentration in commercial batteries [18–21]. Recently, the CSI technique has achieved two-dimensional imaging with a combination of a pinhole and a two-dimensional X-ray detector, which enables high-speed imaging of a two-dimensional cross section of an object [22]. The advantage of CSI over X-ray CT is that CSI has direct access to such two-dimensional cross sections without sample rotation. This is beneficial to the experiments of commercial electrochemical cells in operation. High-energy X-ray radiography is not sensitive to light element substances, such as liquid water and carbon materials, since the X-ray absorption is substantially reduced for such light elements at high energies. On the other hand, X-ray Compton scattering is sensitive to light-element materials [23,24]. It is also noted here that the radiation damage is small since the photoelectric absorption is substantially reduced in high-energy X-rays.

A drawback of the CSI technique is its low data-accumulation efficiency due to the small aperture size of the pinhole. This will be overcome by adopting a system of coded aperture masks [25].

In this study, we have performed a feasibility test of synchrotron-based CSI for liquid water observation in fuel cells. This technique is applied to a material of gas diffusion layer (GDL) in dry or wet, to probe the effect of water penetration into the GDL material.

## 2. Materials and Methods

The CSI experiment was performed with the high-energy inelastic scattering beamline, BL08W, of SPring-8. The experimental setup is shown in Figure 1. The energy of the incident X-rays was 115 keV. The incident X-ray beams were focused to 10 μm in the vertical direction using a Ni compound refractive lens [26,27] and the horizontal direction was 1 mm in the asymmetric Johan monochromator [28]. Compton scattered X-rays from the GDL material, dry or wet, were detected by the two-dimensional CdTe X-ray detector with a pixel size of 25 μm and a pixel array of 80 × 80 (HEXITEC) at a scattering angle of 90 degrees, through a W pinhole with a diameter of 50 μm. A cross-sectional image was taken with an in-plane spatial resolution of 75 μm. The energy spectra of the Compton scattered X-rays were measured at each CdTe pixel. The measurement time was 300 s for each image. From the measured X-ray spectra, the Compton scattering intensity was evaluated by integrating the spectra over the energy range of the Compton scattered X-rays.

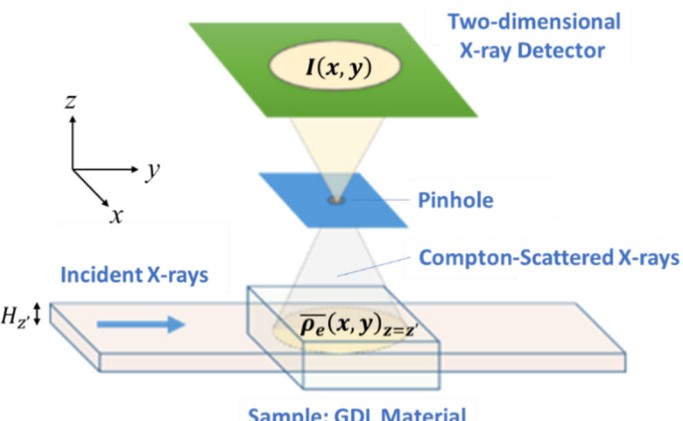

**Figure 1.** Experimental setup for Compton scattering imaging.

The intensity of the Compton scattered X-rays is proportional to the effective electron density at the gauge volume. The intensity distribution on the 2D X-ray detector, $I(x, y)$, is given by the following:

$$I(x,y) \propto \int_{z'-\frac{H_{z'}}{2}}^{z'+\frac{H_{z'}}{2}} \rho_e(x,y,z)dz = \overline{\rho_e}(x,y)_{z=z'} H_{z'} \tag{1}$$

where $\rho_e(x, y, z)$ is three-dimensional electron density; $H_{z'}$ the vertical width (z-direction) of the incident X-ray beams; $\overline{\rho_e}(x, y)_{z=z'}$ is the two-dimensional, average electron density at the vertical position $z'$, integrated over the width of $H_{z'}$. In this experiment, $H_{z'}$ was 10 μm.

The GDL material used in this study was a porous carbon fiber and carbon composite with a Teflon treatment, TGP-H-030 (TORAY Industries, Inc., Tokyo, Japan). The size was 30 mm × 30 mm, and the thickness was 110 μm. As shown in Figure 2, a copper plate with a thickness of 500 μm was placed on the GDL material, dry or wet.

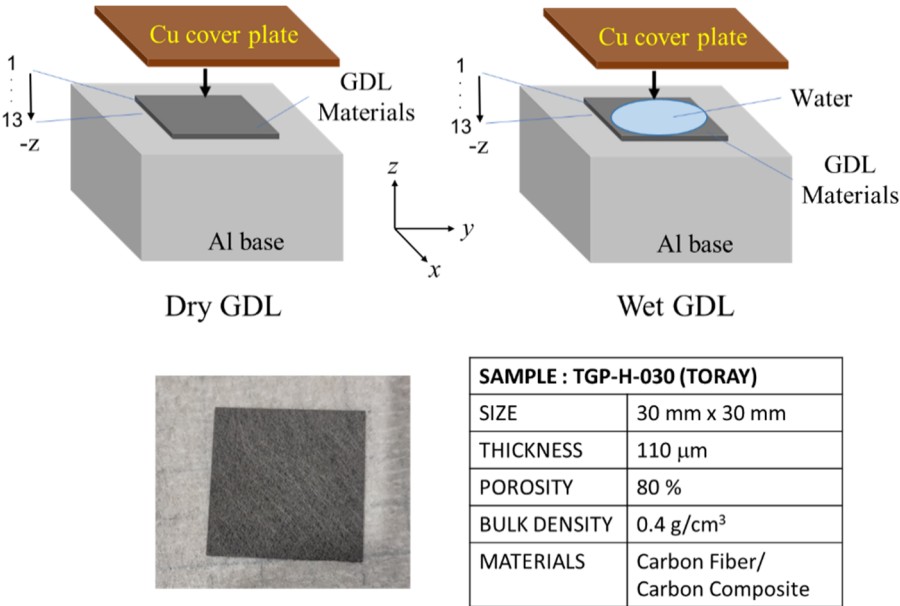

| SAMPLE : TGP-H-030 (TORAY) | |
|---|---|
| SIZE | 30 mm x 30 mm |
| THICKNESS | 110 μm |
| POROSITY | 80 % |
| BULK DENSITY | 0.4 g/cm³ |
| MATERIALS | Carbon Fiber/ Carbon Composite |

**Figure 2.** Samples of the GDL (Gas Diffusion Layer) material, dry or wet, used for the Compton scattering imaging.

### 3. Results and Discussion

*3.1. Spatial Resolution*

The spatial resolution of the CSI technique depends on the pixel size of the two-dimensional X-ray detector, the size of the pinhole, the distance between the X-ray detector and the pinhole and that between the pinhole and the sample. In order to evaluate the overall spatial resolution, a combined piece of Cu/Al/Cu plates with a thickness of 100 μm each (see Figure 3a) has been measured with a pinhole whose size is 100 μm or 50 μm. The obtained images are shown in Figure 3c,d, which display three vertical lines corresponding to the three plates. Figure 3b plots the intensity along the line in Figure 3d. The width of the peaks leads to the overall spatial resolution of 75 μm for the 50 μm pinhole.

*3.2. Cross-Sectional Images of the Dry and Wet Samples*

Figure 4 shows the cross-sectional images of the dry GDL material. Incident X-rays come along the *y* direction, and the Compton scattered X-rays toward the *z* direction are detected by the 2D X-ray detector through the 50 μm pinhole. Since the width of the incident X-rays is 1 mm, only the central area along the *x* direction is illuminated. Image 1 corresponds to the gap between the Cu cover plate and the GDL material, and Image

13 is that close to the Al base. Since the intensity of the Compton scattered X-rays probes the electron density, the GDL material (i.e., carbon fiber and carbon composite) is clearly visualized between Image 2 and Image 12. Since the cross-sectional images have been successively taken with a step of 10 µm along the *z* direction, the stacked images construct the three-dimensional structure of the GDL material.

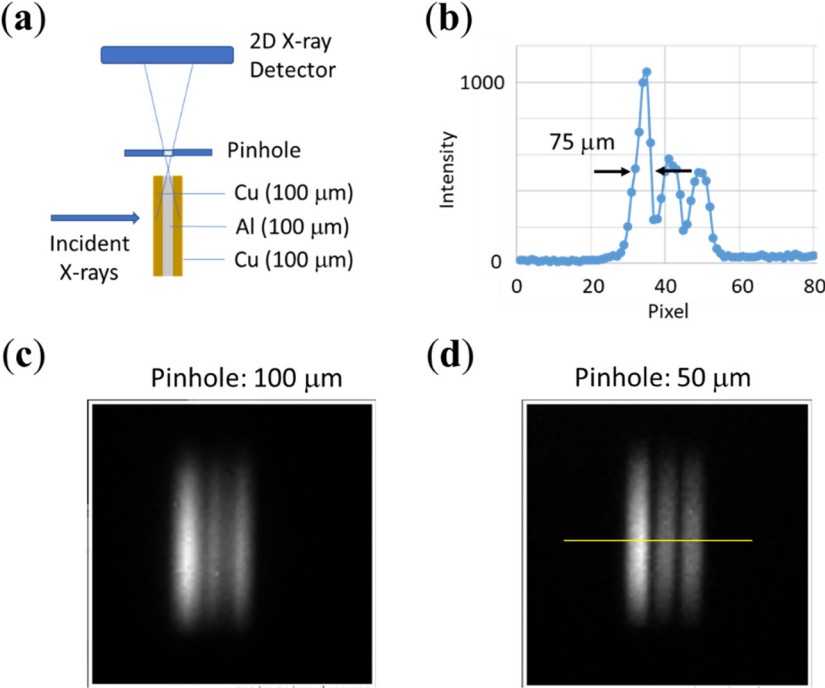

**Figure 3.** Spatial resolution: (**a**) sample and experimental setup; (**b**) intensity along the line in Figure 3d; (**c**) image observed with a 100 µm pinhole; (**d**) image observed with a 50 µm pinhole.

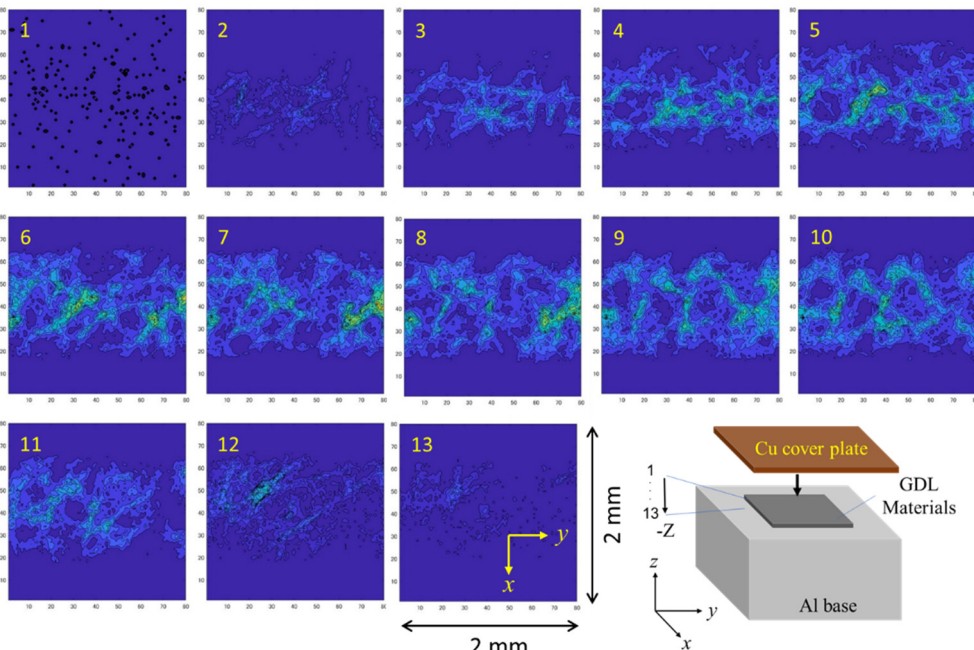

**Figure 4.** Cross-sectional images of dry GDL material. Incident X-rays come along the *y* direction and the Compton scattered X-rays toward the *z* direction are detected by a 2D X-ray detector through a 50 µm pinhole.

Figure 5 shows the cross-sectional images of the wet GDL material, taken under the same conditions as for the dry GDL material. Liquid water is provided from the upper side, and then the Cu cover plate is placed on the water-containing sample. As shown in Images 1–3, liquid water fills the gap between the GDL material and the Cu cover plate, and the water content decreases as the cross section moves away from the Cu cover plate since the GDL material is hydrophobic.

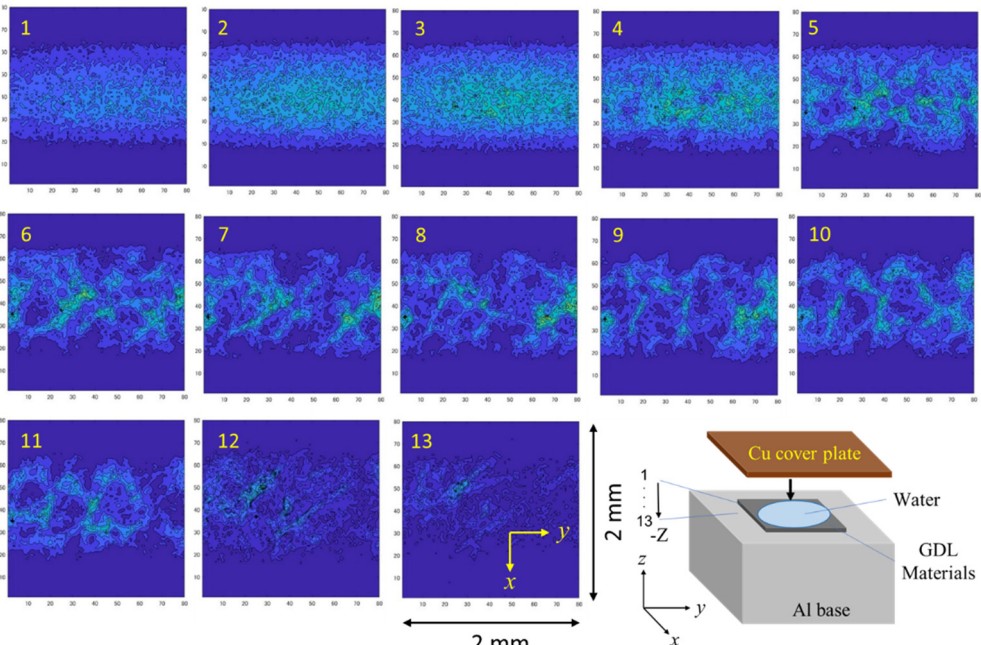

**Figure 5.** Cross-sectional images of wet GDL material. Incident X-rays come along the $y$ direction and the Compton scattered X-rays toward the $z$ direction are detected by a 2D X-ray detector through a 50 μm pinhole.

### 3.3. Imaging Liquid Water in GDL Material

In order to visualize the liquid water distribution in the GDL material, the difference in the images between the dry and wet samples has been examined for the cross-sectional Image 5. Here we assume that the both images are already normalized since the data-accumulation time is the same. As shown in Figure 6c, the difference image (Wet – Dry) successfully shows penetrating liquid water. The areas defined by the yellow lines correspond to the pore spaces in the dry sample (Dry), indicating that liquid water penetrates into the GDL material along the carbon fiber and carbon composite. In addition, the difference image reveals the negative contributions on the region of the carbon fiber and carbon composite, showing possible interactions between the GDL material and liquid water. The details need to be elucidated by further studies.

Figure 7 shows the distribution of the liquid water in cross-sections 5–7 (upper figures), together with that of the carbon fiber and carbon composite (lower figures). One can notice that the water droplets shrink as it separates from the upper surface of the GDL material.

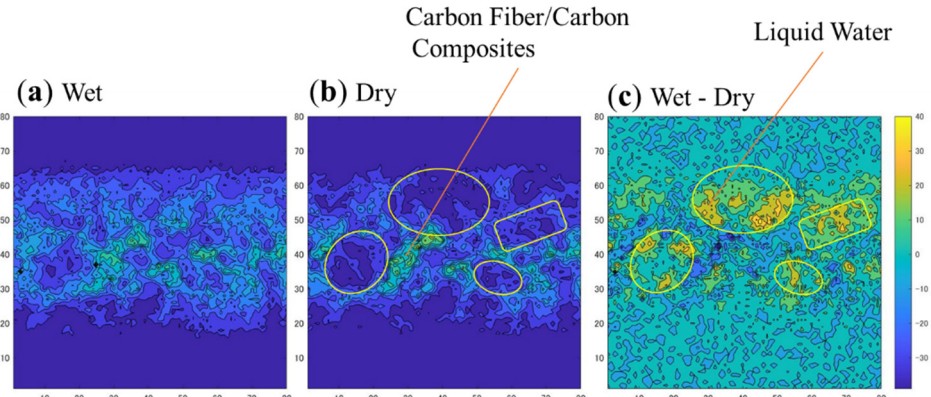

**Figure 6.** Imaging liquid water in cross-section 5: (**a**) wet GDL material sample; (**b**) dry GDL material sample; (**c**) difference image between the wet and dry samples. In (**c**), liquid water is observed in the pore spaces in the GDL material sample, which is indicated by the yellow lines. Negative contributions are observed on the region of the carbon fibers, indicating interactions between the GDL materials and liquid water.

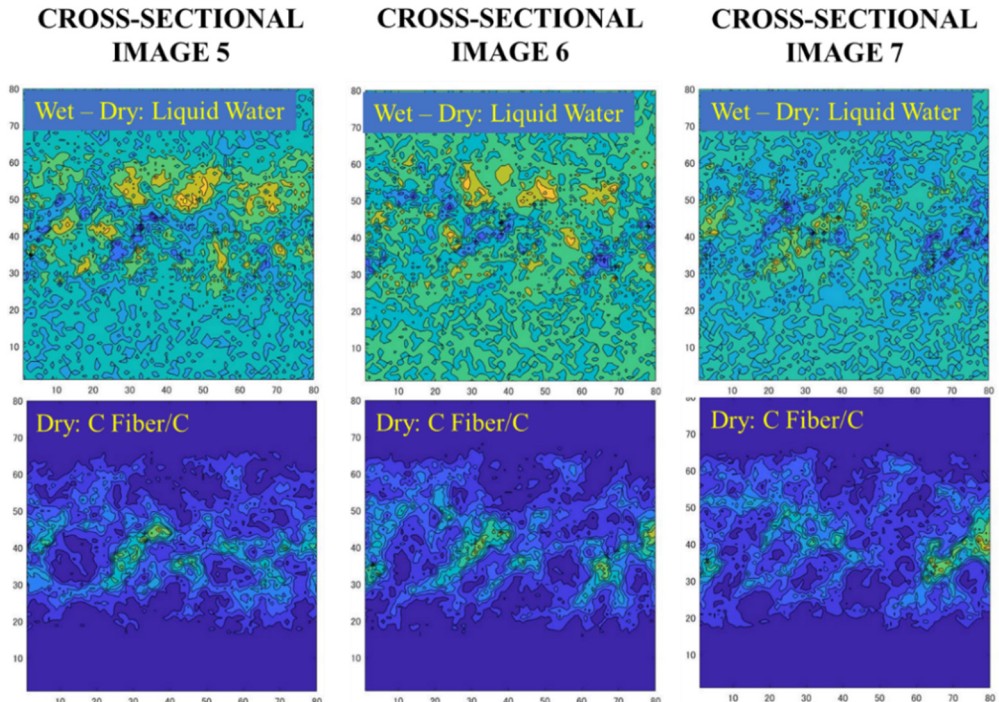

**Figure 7.** Imaging liquid water in cross-sections 5–7, together with the image of the dry sample.

### 3.4. Relative Volume Fraction of C Fiber/C Composites and Liquid Water

To better understand the liquid water distribution along the *z* direction, the variation in the relative volume fraction of the liquid water was evaluated. From Equation (1), the intensity of the Compton scattered X-rays, integrated over a cross section, is given by

$$I \propto F_C \cdot \rho_{e,C} + F_{H_2O} \cdot \rho_{e,H_2O} \tag{2}$$

where $F_C$ and $F_{H_2O}$ are the volume fraction of carbon fiber and carbon composite and that of liquid water, averaged over the cross section, and $\rho_C$ and $\rho_{H_2O}$ are the effective electron density of the carbon fiber and carbon composite and the liquid water, respectively.

Assuming that the porosity of the GDL material is 80% and the density of the carbon composite is 1.8 g/cm$^3$, Equation (2) is approximated by

$$I \propto 18 + 0.6x \tag{3}$$

where $x$ is the volume fraction (%) of liquid water.

Figure 8 shows the depth dependency of the relative volume fraction of liquid water, together with that of the carbon fiber and carbon composite. Here, the X-ray beam position, Z, at 0 µm corresponds to the cross section nearby the Cu cover plate, and that at 140 µm corresponds to the cross section close to the Al base. This result show that liquid water penetrates into the GDL materials at 50 µm from the surface, which is consistent with the observation of the difference images in Figure 7. Some water penetration from the Al base side is also observed. This water comes from the liquid water that has moved outside the GDL material sample.

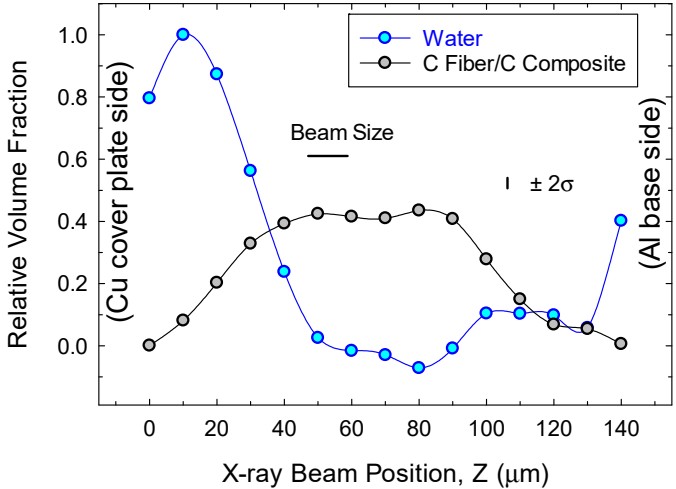

**Figure 8.** The depth dependency of the relative volume fraction of liquid water, together with that of the carbon fiber and carbon composite.

## 4. Conclusions

This study applies the high-energy, synchrotron X-ray CSI technique to the GDL material. We have successfully visualized the liquid water penetrating into the GDL material. The depth-resolved intensity analysis shows that the volume fraction of liquid water decreases monotonically as it separates from the surface, and it penetrates at a depth of 50 µm. These results demonstrate that the imaging water is feasible with the synchrotron-based CSI technique. Finally, since it uses high-energy X-rays with high material penetration and does not require sample rotation, the CSI technique can be applied to visualize the liquid water distribution and its dynamics in a working fuel cell. The challenges are to improve data-accumulation efficiency using a coded aperture system and to improve the spatial resolution using a smaller pinhole aperture.

**Author Contributions:** Conceptualization, Y.U. and Y.S.; methodology, N.T. and Y.S.; software, N.T.; validation, Y.S., Y.U., H.I. and Y.T.; formal analysis, N.T.; investigation, N.T.; resources, Y.S.; data curation, N.T.; writing—original draft preparation, N.T.; writing—review and editing, Y.S.; visualization, N.T. and Y.S.; supervision, H.I.; project administration, Y.T.; funding acquisition, H.I. All authors have read and agreed to the published version of the manuscript.

**Funding:** This research was funded by New Energy and Industrial Technology Development Organization (NEDO), grant number 20001309-0.

**Institutional Review Board Statement:** Not applicable.

**Informed Consent Statement:** Not applicable.

**Data Availability Statement:** The data presented in this study are available on request from the corresponding author.

**Acknowledgments:** Compton scattering imaging was performed with the approval of JASRI (Proposal No. 2020A2148).

**Conflicts of Interest:** The authors declare no conflict of interest.

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
