# Peer review of "Compton Scattering Imaging of Liquid Water in Porous Carbon-Based Materials"

_applsci, doi:10.3390/app11093851_

Round 1

Reviewer 1 Report

Excellent work on this novel/new imaging technique. It is extremely interesting to analyze possible applications of Compton Imaging techniques. Detecting liquid water in fuel cell materials could be very promising, although long exposure times are necessary, what is the main drawback of this technique together with the rather low spatial resolution. However, I like this study very much.

It is important to investigate this young and novel measurement technique   (actually it is very old, but together with new applications it can be said, that it is a new techniques) in order to understand where it could be applied in future. Like neutron imaging it may have certain areas where it is unique and cannot be exchanged by other techniques.

Therefore this study is highly innovative fundamental research that might become important in future once we know how the technique can be used efficiently.

I wish the authors good luck for their future work!

Author Response

Dear Reviewer,

Thank you for your careful reading of our manuscript. Our point-by-point response is in the attachment.

Best regards,

Yoshiharu Sakurai

Reviewer 2 Report

The manuscript describes the “Compton Scattering Imaging of Liquid Water in Porous Carbon-based Materials”. The authors have applied synchrotron-based Compton scattering imaging technique to water contained porous carbon composite and provided the cross-sectional distributions of liquid water, as well as the depth dependency of water content. The objective of this manuscript is quite interesting and the presented results are meaningful. I recommend publishing this paper after addressing following questions.

1) Please carefully correct the units in the manuscript, especially “µm”.

2) How the difference between wet and dry images were made? Why the images became brighter after taking the difference? Authors should add the scale in the images if possible.

3) Ideally the water should not be present near Al-base side, however the relative volume fraction of liquid water is above 0.4. Could the authors justify this? Please add the error bars in Figure 8.

Author Response

Dear Reviewer,

Thank you for your careful reading of our manuscript. Our point-by-point response is in the attachment,

Best regards,

Yoshiharu Sakurai

Reviewer 3 Report

In this manuscript, Compton scattering imaging was used to investigate the water distribution in porous carbon-based materials. This study addresses an important topic since the water management is crucially important for fuel cell operation. There are some questions and suggestions that will make this manuscript more appropriate for publication.      

  1. What was the accelerating voltage used for Compton scattering imaging in this manuscript? The Nafion membrane used in fuel cell would be damaged at high accelerating voltage.

  1. It would be interesting if the Compton scattering imaging could be applied to visualize the liquid water distribution in a working fuel cell. What are the challenges?

  1. Abbreviations should be defined the first time they are used, then the abbreviation is used instead of the full name. Page 1 line 38, please use the abbreviation “CT” for “computed tomography”; Page 2, line 66 and page 7 line 199, please use the abbreviation “CSI” for “Compton scattering imaging”.

  1. Please correct the unit on page 2, lines 68, 73, 74; page 3 lines 90, 94, 107, 110; page 4 lines 116, 127; page 5 lines 138, 143; page 7 lines 187, 189, 203.

  1. Typo: Figure 3: It should be “(b) Intensity along the line in Fig. 3(d)”.

Author Response

(The authors gave the same response as above.)
